# A Cancer-Specific Monoclonal Antibody against HER2 Exerts Antitumor Activities in Human Breast Cancer Xenograft Models

**DOI:** 10.3390/ijms25031941

**Published:** 2024-02-05

**Authors:** Mika K. Kaneko, Hiroyuki Suzuki, Tomokazu Ohishi, Takuro Nakamura, Tomohiro Tanaka, Yukinari Kato

**Affiliations:** 1Department of Antibody Drug Development, Tohoku University Graduate School of Medicine, 2-1 Seiryo-machi, Aoba-ku, Sendai 980-8575, Japan; mika.kaneko.d4@tohoku.ac.jp (M.K.K.); hiroyuki.suzuki.b4@tohoku.ac.jp (H.S.); takuro.nakamura.a2@tohoku.ac.jp (T.N.); tomohiro.tanaka.b5@tohoku.ac.jp (T.T.); 2Institute of Microbial Chemistry (BIKAKEN), Microbial Chemistry Research Foundation, 18-24 Miyamoto, Numazu 410-0301, Japan; ohishit@bikaken.or.jp; 3Laboratory of Oncology, Institute of Microbial Chemistry (BIKAKEN), Microbial Chemistry Research Foundation, 3-14-23 Kamiosaki, Shinagawa-ku, Tokyo 141-0021, Japan

**Keywords:** HER2, cancer-specific monoclonal antibody, epitope, xenograft, breast cancer

## Abstract

Monoclonal antibody (mAb)-based and/or cell-based immunotherapies provide innovative approaches to cancer treatments. However, safety concerns over targeting normal cells expressing reactive antigens still exist. Therefore, the development of cancer-specific mAbs (CasMabs) that recognize cancer-specific antigens with in vivo antitumor efficacy is required to minimize the adverse effects. We previously screened anti-human epidermal growth factor receptor 2 (HER2) mAbs and successfully established a cancer-specific anti-HER2 mAb, H_2_Mab-250/H_2_CasMab-2 (IgG_1_, kappa). In this study, we showed that H_2_Mab-250 reacted with HER2-positive breast cancer cells but did not show reactivity to normal epithelial cells in flow cytometry. In contrast, a clinically approved anti-HER2 mAb, trastuzumab, recognized both breast cancer and normal epithelial cells. We further compared the affinity, effector activation, and antitumor effect of H_2_Mab-250 with trastuzumab. The results showed that H_2_Mab-250 exerted a comparable antitumor effect with trastuzumab in the mouse xenograft models of BT-474 and SK-BR-3, although H_2_Mab-250 possessed a lower affinity and effector activation than trastuzumab in vitro. H_2_Mab-250 could contribute to the development of chimeric antigen receptor-T or antibody–drug conjugates without adverse effects for breast cancer therapy.

## 1. Introduction

The overexpression of human epidermal growth factor receptor 2 (HER2) is observed in approximately 20% of breast cancers [1] and 20% of gastric cancers [2], which are associated with higher rates of recurrence and shorter overall survival. HER2 forms heterodimers with other HER members and the ligands or ligand-independent homodimers when overexpressed [3]. The formation of hetero- or homodimers leads to the activation of downstream signaling such as RAS-ERK and PI3K-AKT pathways, which promote cancer cell proliferation, survival, and invasiveness [3]. A clinically approved anti-HER2 monoclonal antibody (mAb), trastuzumab, showed an anti-proliferative effect in vitro and a potent antitumor efficacy in vivo [4,5]. In the treatment of breast cancer patients with metastasis, trastuzumab is administered in patients with HER2-overexpressed tumors, which are defined by strong and complete membranous staining of more than 10% of cells in immunohistochemistry (IHC 3+) and/or in situ hybridization (ISH)-amplified [6]. The combination therapy of chemotherapeutic agents with trastuzumab improves objective response rates, progression-free survival, and overall survival in HER2-positive breast cancer patients with metastasis [7]. Therefore, trastuzumab has become the most effective therapy for HER2-positive breast cancers [8] and HER2-positive gastric cancers [9].

The trastuzumab-based antibody–drug conjugates (ADCs) such as trastuzumab-deruxtecan (T-DXd) have been evaluated in various clinical trials [10]. Based on the studies, T-DXd has been approved in not only HER2-positive breast cancer [11,12] but also HER2-mutant lung cancer [13] and HER2-low (IHC 1+ or IHC 2+/ISH-non-amplified) advanced breast cancer [14]. Since approximately half of all breast cancers are classifiable as HER2-low, a significant number of patients can benefit from T-DXd therapy [15].

Anti-HER2 therapeutic mAbs and the ADCs have common adverse effects such as cardiotoxicity [16]. Patients must receive routine cardiac monitoring [17]. Moreover, *ErbB2* (ortholog of *HER2*)-knockout mice showed embryonic lethal phenotype because of the lack of cardiac trabeculae [18]. The *ErbB2*-conditional knockout mice in the ventricular area displayed the features of dilated cardiomyopathy [19]. These results indicate that HER2 is essential for normal heart development and homeostasis. Therefore, more selective anti-HER2 mAbs against cancers are necessary to reduce heart failures.

We previously developed 278 clones of anti-HER2 mAbs using recombinant HER2 ectodomain (HER2ec) derived from glioblastoma LN229 [20] or HER2-overexpressed LN229 [21] as antigens. We further screened the reactivity to HER2-positive breast cancers (BT-474 and SK-BR-3) and normal epithelial cells using flow cytometry. Finally, we successfully developed a cancer-specific anti-HER2 mAb, H_2_Mab-250/H_2_CasMab-2 (IgG_1_, kappa) [22]. Importantly, H_2_Mab-250 did not react with non-transformed normal epithelial cells (HaCaT and MCF 10A) and immortalized normal epithelial cells derived from the mammary gland, gingiva, lung bronchus, corneal, thymus, kidney proximal tubule, and colon [22]. In contrast, most anti-HER2 mAbs including trastuzumab reacted with both cancer and normal epithelial cells [22]. Furthermore, the results of IHC revealed that H_2_Mab-250 possesses high reactivity to the HER2-positive breast cancer tissues and did not react with normal tissues, including the heart [22]. The epitope mapping demonstrated that the Trp614 in HER2 domain IV mainly contributes to the recognition by H_2_Mab-250 [22].

Trastuzumab is a humanized IgG_1_ mAb that binds to Fcγ receptors (FcγRs) on various immune cells [23]. The FcγR binding activates macrophages, dendritic cells, and neutrophils, which change adaptive immune responses by antigen presentation, cytokine production, and chemotaxis [4]. Moreover, the FcγR engagement activates natural killer (NK) cells and macrophages, which can result in the target cell lysis, termed antibody-dependent cellular cytotoxicity (ADCC) [4]. In mouse mAbs, IgG_2a_ or IgG_2b_ can bind to FcγR with high affinity [24]. Furthermore, a core fucose deficiency on the Fc *N*-glycan has been shown to enhance the binding of IgG to FcγR on effector cells [25] and exert potent antitumor effects [26]. The defucosylated recombinant mAbs can be produced using fucosyltransferase 8-knockout Chinese hamster ovary (CHO) cells [27].

In this study, we produced a mouse IgG_2a_-type mAb and a human IgG_1_-type mAb from H_2_Mab-250. We then compared the affinity, effector activation, and antitumor effect of H_2_Mab-250 with trastuzumab.

## 2. Results

### 2.1. The Binding Affinity of H_2_Mab-250 and Trastuzumab

H_2_Mab-250 recognized HER2 expressed in breast cancers (BT-474 and SK-BR-3), but it did not recognize HER2 in normal epithelial cells, although trastuzumab recognized both types of HER2 [22]. Since H_2_Mab-250 and trastuzumab are mouse IgG_1_ and human IgG_1_, respectively, we generated the same isotype of recombinant mAbs (mouse IgG_2a_ or human IgG_1_), which possess ADCC, to compare the antitumor activities between H_2_Mab-250 and trastuzumab. H_2_Mab-250-mG_2a_ and tras-mG_2a_ are mouse IgG_2a_-type mAbs derived from H_2_Mab-250 and trastuzumab, respectively. H_2_Mab-250-hG_1_ and trastuzumab are human IgG_1_-type mAbs.

We confirmed the reactivity of the mAbs against breast cancer cells and normal epithelial cells. As shown in Figure 1A, H_2_Mab-250-mG_2a_ reacted with HER2-overexpressed CHO-K1 (CHO/HER2), HER2-positive BT-474, and SK-BR-3 cells, but not with triple-negative MDA-MB-468 cells. Importantly, H_2_Mab-250-mG_2a_ did not react with spontaneously immortalized normal epithelial cells (HaCaT (keratinocyte) and MCF 10A (mammary gland)) and immortalized normal epithelial cells (HBEC3-KT (lung bronchus), hTERT-HME1 (mammary gland), hTCEpi (corneal), hTEC/SVTERT24-B (thymus), RPTEC/TERT1 (kidney proximal tubule), and HCEC-1CT (colon)). In contrast, tras-mG_2a_ showed the reactivity to both breast cancer and normal epithelial cells. Similar reactivities were observed in H_2_Mab-250-hG_1_ and trastuzumab (Figure 1B). These results indicated that H_2_Mab-250-mG_2a_ and H_2_Mab-250-hG_1_ retain the cancer specificity compared with tras-mG_2a_ and trastuzumab, respectively.

We next evaluated the binding affinity of H_2_Mab-250-mG_2a_, H_2_Mab-250-hG_1_, tras-mG_2a_, and trastuzumab to HER2ec by enzyme-linked immunosorbent assay (ELISA). The *K*_D_ values of H_2_Mab-250-mG_2a_ and tras-mG_2a_ to HER2ec were determined to be 1.1 × 10^−9^ M (Figure 2A) and 1.9 × 10^−10^ M (Figure 2B), respectively. The *K*_D_ values of H_2_Mab-250-hG_1_ and trastuzumab to HER2ec were also determined to be 1.5 × 10^−9^ M (Figure 2C) and 2.0 × 10^−10^ M (Figure 2D), respectively. Although the binding affinity of H_2_Mab-250-mG_2a_ and H_2_Mab-250-hG_1_ was less than that of tras-mG_2a_ and trastuzumab, respectively, H_2_Mab-250-mG_2a_ and H_2_Mab-250-hG_1_ exhibited high binding affinity to HER2ec. Furthermore, the changes in the mAb format did not affect the binding affinity compared to the original mAbs [22].

### 2.2. The Ability of Effector Cell Activation by the Derivatives of H_2_Mab-250 and Trastuzumab

The mAb-FcγRIIIa binding-mediated ADCC pathway activation in effector cells can be quantified by a bioluminescent reporter gene assay, called the ADCC reporter bioassay [28]. We previously showed that H_2_Mab-250-mG_2a_ selectively activates the effector cells against breast cancer cells, but not against normal cells. In contrast, trastuzumab activated the effector with a similar 50% effective concentration (EC_50_) against breast cancer and normal cells [22]. We next examined whether the derivatives of H_2_Mab-250 and trastuzumab could activate the ADCC pathway in the presence of BT-474 and SK-BR-3 cells. To compare the ADCC pathway activation by the derivatives of H_2_Mab-250 and trastuzumab, we treated BT-474 and SK-BR-3 cells with serially diluted mAbs, and then we incubated them with effector Jurkat cells, which express the human FcγRIIIa, and a firefly luciferase reporter gene driven by a nuclear factor of activated T cell (NFAT)-responsive element. As shown in Figure 3A, H_2_Mab-250-mG_2a_ could activate the effector (EC_50_: 1.6 × 10^−5^ g/mL), but it was less effective than tras-mG_2a_ (EC_50_: 3.7 × 10^−8^ g/mL) in the presence of BT-474 cells. H_2_Mab-250-mG_2a_ and tras-mG_2a_ also exhibited a similar relationship to EC_50_ (3.9 × 10^−6^ g/mL and 9.0 × 10^−9^ g/mL, respectively) in SK-BR-3 cells (Figure 3B). Furthermore, H_2_Mab-250-hG_1_ could activate the effector (EC_50_: 1.4 × 10^−6^ g/mL), but it was less effective than trastuzumab (EC_50_: 1.5 × 10^−8^ g/mL) in BT-474 cells (Figure 3A). H_2_Mab-250-hG_1_ and trastuzumab also exhibited a similar relationship to EC_50_ (2.2 × 10^−7^ g/mL and 1.5 × 10^−9^ g/mL, respectively) in SK-BR-3 cells (Figure 3B). These results indicated that H_2_Mab-250-mG_2a_ and H_2_Mab-250-hG_1_ possess a lower ability of effector cell activation than tras-mG_2a_ and trastuzumab, respectively.

### 2.3. Immunohistochemical Analysis by H_2_Mab-250 and Trastuzumab in Breast Cancer Tissue

We previously showed that H_2_Mab-250 could stain the HER2-positive breast cancer tissue, but not normal tissues, including the heart, breast, stomach, lung, colon, kidney, and esophagus in IHC [22]. We next compared the reactivity of H_2_Mab-250 with tras-mG_2a_ using a formalin-fixed paraffin-embedded (FFPE) tissue of HER2-positive breast cancer. In contrast to the binding affinity, H_2_Mab-250 exhibited superior reactivity to the breast cancer cells over tras-mG_2a_ (Figure 4). H_2_Mab-250 showed clear tumor staining even at 1/20 concentration of tras-mG_2a_ (0.5 µg/mL) (Figure 4A). We also confirmed that H_2_Mab-250 did not stain normal breast and heart tissues (Figure 4B).

### 2.4. Antitumor Activities by H_2_Mab-250-mG_2a_ and Tras-mG_2a_

We next investigated the in vivo antitumor effect using H_2_Mab-250-mG_2a_ and tras-mG_2a_ in the BT-474 and SK-BR-3 xenograft models. We injected H_2_Mab-250-mG_2a_, tras-mG_2a_, and a control mouse mAb (PMab-231) intraperitoneally on days 6, 13, and 20 after BT-474 and SK-BR-3 inoculation. We measured the tumor volume on days 6, 13, 20, and 27 following the inoculation. The H_2_Mab-250-mG_2a_ administration led to a significant reduction in BT-474 and SK-BR-3 xenograft on days 20 (*p* < 0.01) and 27 (*p* < 0.01) compared with that of the control (Figure 5A,B). In contrast, the tras-mG_2a_ administration also led to a significant reduction in BT-474 and SK-BR-3 xenograft on days 20 (*p* < 0.01) and 27 (*p* < 0.01) compared with that of the control (Figure 5A,B). The H_2_Mab-250-mG_2a_ and tras-mG_2a_ administration resulted in 58% and 56% reductions (BT-474) and 53% and 50% reductions (SK-BR-3) of tumor volume compared with that of the control mAb (PMab-231) on day 27, respectively.

The BT-474 tumors from the H_2_Mab-250-mG_2a_-treated and tras-mG_2a_-treated mice weighed significantly less than those from the control mouse mAb-treated mice (71% and 67% reduction, respectively; *p* < 0.01, Figure 5C,E). The SK-BR-3 tumors from the H_2_Mab-250-mG_2a_-treated and tras-mG_2a_-treated mice weighed significantly less than those from mouse control mAb-treated mice (53% and 46% reduction, respectively; *p* < 0.01, Figure 5D,F). There was no significant difference between H_2_Mab-250-mG_2a_-treated and tras-mG_2a_-treated tumors.

The body weight loss was slightly observed on days 6 and 13 in H_2_Mab-250-mG_2a_-treated BT-474 xenograft-bearing mice (Figure 5G), but there was no significant difference in SK-BR-3 xenograft-bearing mice (Figure 5H).

### 2.5. Antitumor Activities by H_2_Mab-250-hG_1_ and Trastuzumab

We next evaluated the antitumor activity of H_2_Mab-250-hG_1_ and trastuzumab in the BT-474 and SK-BR-3 xenograft models. We injected H_2_Mab-250-hG_1_ and trastuzumab, as well as control human IgG, intraperitoneally on days 7, 14, and 21 after BT-474 and SK-BR-3 inoculation. Furthermore, human NK cells were injected around the tumors on the same days of the mAb injection. We measured the tumor volume on days 7, 14, 21, and 28 following the inoculation. The H_2_Mab-250-hG_1_ administration led to a significant reduction in BT-474 and SK-BR-3 xenograft on days 14 (*p* < 0.05 (BT-474), *p* < 0.01 (SK-BR-3)), 21 (*p* < 0.01), and 28 (*p* < 0.01) compared with that of the control (Figure 6A,B). In contrast, the trastuzumab administration also led to a significant reduction in BT-474 and SK-BR-3 xenograft on days 14 (*p* < 0.01 (BT-474), *p* < 0.05 (SK-BR-3)), 21 (*p* < 0.01), and 28 (*p* < 0.01) compared with that of the control (Figure 6A,B). The H_2_Mab-250-hG_1_ and trastuzumab administration resulted in 62% and 62% reductions (BT-474) and 53% and 53% reductions (SK-BR-3) of tumor volume compared with that of the control on day 28, respectively.

The BT-474 tumors from the H_2_Mab-250-hG_1_- and trastuzumab-treated mice weighed significantly less than those from the control human IgG-treated mice (59% and 57% reduction, respectively; *p* < 0.01, Figure 6C,E). The SK-BR-3 tumors from the H_2_Mab-250-hG_1_ and trastuzumab-treated mice weighed significantly less than those from the control human IgG-treated mice (48% and 48% reduction, respectively; *p* < 0.01, Figure 6D,F). There was no significant difference between H_2_Mab-250-hG_1_- and trastuzumab-treated tumors.

The body weight loss was not observed in H_2_Mab-250-hG_1_- and trastuzumab-treated BT-474 and SK-BR-3 xenograft-bearing mice (Figure 6G,H).

## 3. Discussion

Most therapeutic mAbs including anti-HER2 mAbs possess adverse effects, probably due to the recognition of antigens in normal cells [29]. Therefore, tumor-selective or -specific mAbs would minimize the adverse effects. We developed CasMabs against podoplanin (LpMab-2 and LpMab-23 [30]), podocalyxin (PcMab-6 [31]), and HER2 (H_2_Mab-250 [22]) by evaluating the reactivity against cancer and normal cells in flow cytometry. We also showed the in vivo antitumor effect of the recombinant mAbs (mouse IgG_2a_ or human IgG_1_ types) derived from the abovementioned mAbs [30,31,32]. In this study, we evaluated the antitumor effect of H_2_Mab-250-mG_2a_ and H_2_Mab-250-hG_1_, which exhibited a cancer specificity compared to tras-mG_2a_ and trastuzumab, respectively (Figure 1). Although H_2_Mab-250 possesses ≈10-fold lower affinity (Figure 2) and 100~1000-fold lower effect for effector activation than trastuzumab (Figure 3), mouse IgG_2a_-type and human IgG_1_-type H_2_Mab-250 possess comparable antitumor effects with those of trastuzumab (Figure 5 and Figure 6).

The benefit of low-affinity mAbs for therapeutic applications has been discussed. A low-affinity anti-EGFR mAb (*K*_D_: 3.4 × 10^−7^ M) was efficiently taken up by cancer cells but not normal cells, which resulted in sufficient efficacy against tumor cells, but low toxicity against normal keratinocytes [33]. An anti-HER3 mAb, Ab562, possesses ≈10-fold lower affinity (*K*_D_: 2~3 × 10^−8^ M) than patritumab. The ADC AMT-562 exhibited sufficient antitumor effects with minimizing potential toxicity [34]. Since H_2_Mab-250 exhibited no reactivity to normal epithelial cells in flow cytometry (Figure 1) and IHC [22], H_2_Mab-250 or H_2_Mab-250-ADC could exhibit antitumor efficacy with lower side effects.

H_2_Mab-250-mG_2a_ and H_2_Mab-250-hG_1_ could trigger the ADCC activity to BT-474 and SK-BR-3 cells; however, the effects of H_2_Mab-250-mG_2a_ and H_2_Mab-250-hG_1_ were also lower than that of tras-mG_2a_ and trastuzumab, respectively (Figure 3). In contrast, H_2_Mab-250 exhibited a superior reactivity to HER2-positive breast cancer tissue in immunohistochemistry (Figure 4). H_2_Mab-250 also recognized the HER2-positive breast cancer tissue in the absence of antigen retrieval (Appendix A). We previously identified the H_2_Mab-250 epitope as _613-_IWKFP_-617_ in the HER2 domain IV. The epitope sequence is partially included with the wider binding epitope of trastuzumab (residues 579–625) [35]. Furthermore, we identified Trp614 as a central amino acid in the recognition by H_2_Mab-250 [22]. Since the similar in vivo antitumor efficacy of H_2_Mab-250-mG_2a_ and H_2_Mab-250-hG_1_ was shown compared with tras-mG_2a_ and trastuzumab, respectively (Figure 5 and Figure 6), the _613-_IWKFP_-617_ sequence may be highly accessible by H_2_Mab-250 in vivo due to the unknown mechanism. Compared to the in vitro cell culture condition, in vivo tumor cells received various stresses such as hypoxia [36], nutrient deprivation [37], and abnormal redox state [38]. Further studies are required for the influence of those stresses on the recognition by H_2_Mab-250 in cancer cells.

Chimeric antigen receptor (CAR)-T cell therapy against HER2 has been evaluated in clinical trials [15]. CAR-T cells against CD19 have improved outcomes for patients with B-cell lymphoma. However, disease relapse commonly occurs in many patients [39,40]. Although different mechanisms of immune escape have been demonstrated [41], the tumor cells from many relapsed patients did not exhibit the features of immune escape. Recently, additional mechanisms, such as trogocytosis, have been proposed [42]. When CAR-T cells, possessing the high-affinity anti-CD19 FMC63-based CAR, are co-cultured with CD19-positive lymphoma cells, the CAR-T cells strip CD19 from lymphoma cells and incorporate it into their plasma membrane [42]. This is called “trogocytosis”, which results in the emergence of antigen-negative target cells. Furthermore, the CAR-T cells that acquired CD19 by trogocytosis can be killed by the other CD19 CAR-T cells [42]. A promising approach to limit CAR-T cell-mediated trogocytosis is the reduction of CAR affinity [43]. CD19-targeting CAR with ≈40-fold lower affinity (*K*_D_: 1.4 × 10^−8^ M) than the clinically approved FMC63-based CAR (*K*_D_:  3.3 × 10^−10^ M) was developed [44]. The reduced affinity CAR-T cells exhibited higher efficacy and persistence than FMC63-based CAR-T cells in a mouse model [44], as well as robust antitumor efficacy and persistence in two clinical trials [44,45]. These data show that it is possible to significantly limit trogocytosis by reducing CAR affinity while maintaining antitumor activity as well as clinical efficacy. The property of H_2_Mab-250 could contribute to the development of HER2-targeting CAR-T cells (now in a clinical phase I study in the US) by limiting trogocytosis and maintaining cancer specificity.

## 4. Materials and Methods

### 4.1. Cell Culture

CHO-K1, BT-474, SK-BR-3, MDA-MB-468, MCF 10A, HBEC3-KT, hTERT-HME1, and RPTEC/TERT1 were obtained from the American Type Culture Collection (ATCC, Manassas, VA, USA). Human keratinocyte HaCaT was purchased from the Cell Lines Service GmbH (Eppelheim, Germany). hTCEpi, hTEC/SVTERT24-B, and HCEC-1CT were purchased from EVERCYTE (Vienna, Austria).

CHO-K1 and CHO/HER2 were cultured in Roswell Park Memorial Institute (RPMI)-1640 medium (Nacalai Tesque, Inc., Kyoto, Japan), and BT-474, SK-BR-3, MDA-MB-468, and HaCaT were cultured in Dulbecco’s modified Eagle’s medium (DMEM) (Nacalai Tesque, Inc.), supplemented with 10% heat-inactivated fetal bovine serum (FBS; Thermo Fisher Scientific Inc., Waltham, MA, USA), 100 units/mL of penicillin, 100 μg/mL streptomycin, and 0.25 μg/mL amphotericin B (Nacalai Tesque, Inc.). The mammary epithelial cell line MCF 10A was cultured in Mammary Epithelial Cell Basal Medium BulletKit^TM^ (Lonza, Basel, Switzerland) supplemented with 100 ng/mL cholera toxin (Sigma-Aldrich Corp., St. Louis, MO, USA).

Immortalized normal epithelial cell lines were maintained as follows; HBEC3-KT, Airway Epithelial Cell Basal Medium and Bronchial Epithelial Cell Growth Kit (ATCC); hTERT-HME1, Mammary Epithelial Cell Basal Medium BulletKit^TM^ without GA-1000 (Lonza); hTCEpi, KGMTM-2 BulletKit^TM^ (Lonza); hTEC/SVTERT24-B, OptiPRO^TM^ SFM and GlutaMAX^TM^-I (Thermo Fisher Scientific Inc.); RPTEC/TERT1, DMEM/F-12, and hTERT Immortalized RPTEC Growth Kit with supplement A and B (ATCC); and HCEC-1CT, DMEM/M199 (4:1, Thermo Fisher Scientific Inc.), 2% Cosmic Calf Serum (Cytiva, Marlborough, MA, USA), 20 ng/mL hEGF (Sigma-Aldrich Corp.), 10 μg/mL insulin (Sigma-Aldrich Corp.), 2 μg/mL apo-transferrin (Sigma-Aldrich Corp.), 5 nM sodium-selenite (Sigma-Aldrich Corp.), and 1 μg/mL hydrocortisone (Sigma-Aldrich Corp.).

All cell lines were cultured at 37 °C in a humidified atmosphere with 5% CO_2_ and 95% air.

### 4.2. Production of Recombinant mAbs

To generate recombinant H_2_Mab-250, the V_H_ cDNAs and the C_H_ cDNA of mouse IgG_1_ were cloned into the pCAG-Neo vector (FUJIFILM Wako Pure Chemical Corporation, Osaka, Japan). The V_L_ cDNAs and C_L_ cDNA of the mouse kappa light chain were also cloned into the pCAG-Ble vector (FUJIFILM Wako Pure Chemical Corporation). The vectors were transfected into ExpiCHO-S cells using the ExpiCHO Expression System (Thermo Fisher Scientific, Inc.), and Ab-Capcher (ProteNova Co., Ltd., Kagawa, Japan) was used to purify the recombinant H_2_Mab-250.

To generate mouse IgG_2a_-type H_2_Mab-250 (H_2_Mab-250-mG_2a_), we cloned the V_H_ cDNA of H_2_Mab-250 and C_H_ of mouse IgG_2a_ into the pCAG-Ble vector. The mouse kappa light chain vector of H_2_Mab-250 was described above. To generate a mouse IgG_2a_ type of trastuzumab (tras-mG_2a_), the V_H_ cDNA of trastuzumab and the C_H_ cDNA of mouse IgG_2a_ were cloned into the pCAG-Neo vector, and the V_L_ cDNA of trastuzumab and the C_L_ cDNA of mouse kappa light chain were cloned into the pCAG-Ble vector.

To generate the recombinant PMab-231, we cloned heavy and light chains of PMab-231 [46] into the pCAG-Neo and pCAG-Ble vectors, respectively. To generate a mouse-human chimeric mAb (H_2_Mab-250-hG_1_), V_H_ of H_2_Mab-250 and C_H_ of human IgG_1_ were cloned into the pCAG-Ble vector. The V_L_ of H_2_Mab-250 and C_L_ of the human kappa light chain were cloned into the pCAG-Neo vector. The vectors were transfected into BINDS-09 (fucosyltransferase 8-knockout ExpiCHO-S) cells using the ExpiCHO Expression System (Thermo Fisher Scientific, Inc.). H_2_Mab-250-mG_2a_, tras-mG_2a_, H_2_Mab-250-hG_1_, trastuzumab, and PMab-231 were purified using Ab-Capcher.

Normal human IgG was purchased from Sigma-Aldrich Corp.

### 4.3. Flow Cytometry

Cells were collected using 0.25% trypsin and 1 mM ethylenediamine tetraacetic acid (EDTA; Nacalai Tesque, Inc.). The cells (1 × 10^5^ cells/sample) were treated with primary mAbs (10 μg/mL) or blocking buffer (control; 0.1% bovine serum albumin (BSA) in phosphate-buffered saline (PBS)) for 30 min at 4 °C. Next, the cells were treated with Alexa Fluor 488-conjugated anti-mouse IgG (1:1000; Cell Signaling Technology, Danvers, MA, USA) or FITC-conjugated anti-human IgG (1:1000; Sigma-Aldrich Corp.) for 30 min at 4 °C. The fluorescence data were collected using an SA3800 Cell Analyzer (Sony Corp., Tokyo, Japan), and the data were analyzed using FlowJo v10.8.1 (BD Biosciences, Franklin Lakes, NJ, USA).

### 4.4. ELISA

HER2ec [22] was immobilized on Nunc Maxisorp 96-well immunoplates (Thermo Fisher Scientific Inc.) at a concentration of 1 µg/mL for 30 min at 37 °C. After washing with PBS containing 0.05% (*v*/*v*) Tween 20 (PBST; Nacalai Tesque, Inc.), wells were blocked with 1% (*w*/*v*) BSA-containing PBST for 30 min at 37 °C. The serially diluted H_2_Mab-250, H_2_Mab-250-mG_2a_, H_2_Mab-250-hG_1_, tras-mG_2a_, and trastuzumab (0.0006–10 µg/mL) were added to each well, followed by peroxidase-conjugated anti-mouse immunoglobulins (1:3000 diluted; Agilent Technologies Inc., Santa Clara, CA, USA) or peroxidase-conjugated anti-human immunoglobulins (1:3000 diluted; Sigma-Aldrich Corp.). Enzymatic reactions were conducted using an ELISA POD Substrate TMB Kit (Nacalai Tesque, Inc.), followed by the measurement of the optical density at 655 nm, using an iMark microplate reader (Bio-Rad Laboratories, Inc., Berkeley, CA, USA). The binding isotherms were fitted into the built-in, one-site binding model in GraphPad PRISM 6 (GraphPad Software, Inc., La Jolla, CA, USA) to calculate the dissociation constant (*K*_D_).

### 4.5. ADCC Reporter Bioassay

The ADCC reporter bioassay was performed using an ADCC Reporter Bioassay kit (Promega Corporation, Madison, WI, USA), according to the manufacturer’s instructions. Target cells (BT-474 and SK-BR-3, 12,500 cells per well) were cultured in a 96-well white solid plate. H_2_Mab-250-mG_2a_, H_2_Mab-250-hG_1_, tras-mG_2a_, and trastuzumab were serially diluted and added to the target cells. Jurkat cells stably expressing the human FcγRIIIa receptor and a NFAT-response element driving firefly luciferase were used as effector cells. The engineered Jurkat cells (75,000 cells in 25 μL) were then added and co-cultured with antibody-treated target cells at 37 °C for 6 h. Luminescence using the Bio-Glo Luciferase Assay System was measured using a GloMax luminometer (Promega Corporation).

### 4.6. Immunohistochemical Analysis

FFPE tissue of HER2-positive breast cancer was obtained from the Sendai Medical Center [20]. Informed consent for sample procurement and subsequent data analyses was obtained from the patient or the patient’s guardian at the Sendai Medical Center. A normal tissue array (B901064) was purchased from BioChain Institute Inc. (Eureka Drive, Newark, CA, USA). The antigen retrieval was performed by autoclave in citrate buffer (pH 6.0; Nichirei Biosciences, Inc., Tokyo, Japan) for 20 min. The blocking was performed using SuperBlock T20 (Thermo Fisher Scientific Inc.). The sections were incubated with H_2_Mab-250 (10, 1, 0.5 or 0.1 μg/mL) and tras-mG_2a_ (10 μg/mL) and then treated with the EnVision+ Kit for mouse (Agilent Technologies, Inc., Santa Clara, CA, USA). The chromogenic reaction was performed using 3,3′-diaminobenzidine tetrahydrochloride (DAB; Agilent Technologies, Inc.). Counterstaining was performed using hematoxylin (FUJIFILM Wako Pure Chemical Corporation), and Leica DMD108 (Leica Microsystems GmbH, Wetzlar, Germany) was used to obtain images and examine the sections.

### 4.7. Antitumor Activities of H_2_Mab-250-mG_2a_, Tras-mG_2a_, H_2_Mab-250-hG_1_, and Trastuzumab in Breast Cancer Xenograft Models

To examine the antitumor effect of H_2_Mab-250-mG_2a_ tras-mG_2a_, H_2_Mab-250-hG_1_, and trastuzumab, animal experiments were approved by the Institutional Committee for Experiments of the Institute of Microbial Chemistry (approval no. 2023-060 or 2023-066). During the experimental period, we monitored mice maintained in a pathogen-free environment on an 11 h light/13 h dark cycle with food and water supplied ad libitum. Mice were monitored for health and weight every one or five days. We identified body weight loss exceeding 25% and maximum tumor size exceeding 3000 mm^3^ as humane endpoints and terminated the experiments.

BT-474 and SK-BR-3 cells were suspended in 0.3 mL of 1.33 × 10^8^ cells/mL using DMEM and mixed with 0.5 mL of BD Matrigel Matrix Growth Factor Reduced (BD Biosciences, San Jose, CA, USA). Then, BALB/c nude mice (Jackson Laboratory Japan, Kanagawa, Japan) were injected subcutaneously in the left flank with 100 μL of the suspension (5 × 10^6^ cells). On day 6 post-inoculation, 100 μg of H_2_Mab-250-mG_2a_ (*n* = 8), tras-mG_2a_ (*n* = 8), or control (PMab-231; *n* = 8) in 100 µL PBS was intraperitoneally injected. On days 13 and 20, additional antibody injections were performed. The tumor diameter was measured on days 6, 13, 20, and 27 after the inoculation of cells.

For the evaluation of H_2_Mab-250-hG_1_ and trastuzumab, we injected the mice with 100 μg of H_2_Mab-250-hG_1_ (*n* = 8), trastuzumab (*n* = 8), and control human IgG (*n* = 8) in 100 μL of PBS through intraperitoneal injection on day 7 post-inoculation. Additional antibodies were injected on days 14 and 21. Furthermore, human NK cells (8.0 × 10^5^ cells, Takara Bio, Inc., Shiga, Japan) were injected around the tumors on days 7, 14, and 21. The tumor diameter was measured on days 7, 14, 21, and 28 after inoculation with cells.

The tumor volume was calculated using the following formula: volume = W^2^ × L/2, where W is the short diameter and L is the long diameter. All mice were euthanized by cervical dislocation.

## 5. Conclusions

A cancer-specific anti-HER2 mAb, H_2_Mab-250, exhibited antitumor efficacy in vivo. In the future, H_2_Mab-250 could contribute to the development of CAR-T or ADCs without adverse effects for breast cancer therapy.

## Figures and Tables

**Figure 1 ijms-25-01941-f001:**
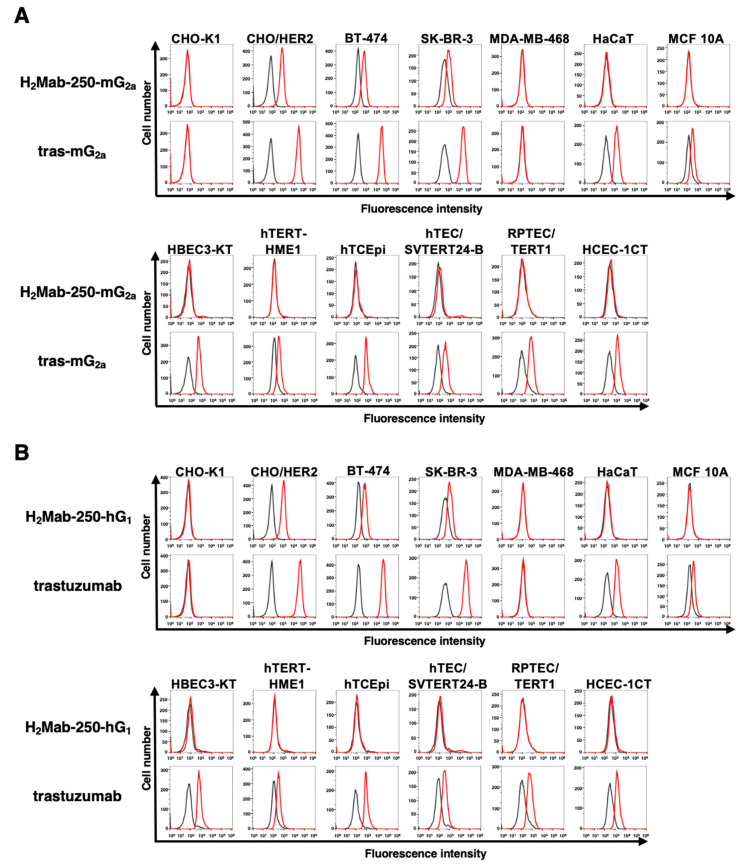
Flow cytometry using anti-HER2 mAbs against HER2-overexpressed cells, breast cancer cell lines, and normal epithelial cell lines. (**A**) Flow cytometry using H_2_Mab-250-mG_2a_ (10 μg/mL; red line) and tras-mG_2a_ (10 μg/mL; red line) against CHO-K1, CHO/HER2, HER2-positive breast cancers (BT-474 and SK-BR-3), a triple-negative breast cancer (MDA-MB-468), and spontaneously immortalized normal epithelial cells (HaCaT and MCF 10A), as well as immortalized normal epithelial cells including HBEC3-KT (lung bronchus), hTERT-HME1 (mammary gland), hTCEpi (corneal), hTEC/SVTERT24-B (thymus), RPTEC/TERT1 (kidney proximal tubule), and HCEC-1CT (colon). (**B**) Flow cytometry using H_2_Mab-250-hG_1_ (10 μg/mL; red line) and trastuzumab (10 μg/mL; red line) against the abovementioned cell lines. The cells were treated with Alexa Fluor 488-conjugated anti-mouse IgG or fluorescein (FITC)-conjugated anti-human IgG. The fluorescence data were collected using an SA3800 Cell Analyzer (Sony Corp., Tokyo, Japan), and the data were analyzed using FlowJo v10.8.1. The black line represents the negative control (blocking buffer).

**Figure 2 ijms-25-01941-f002:**
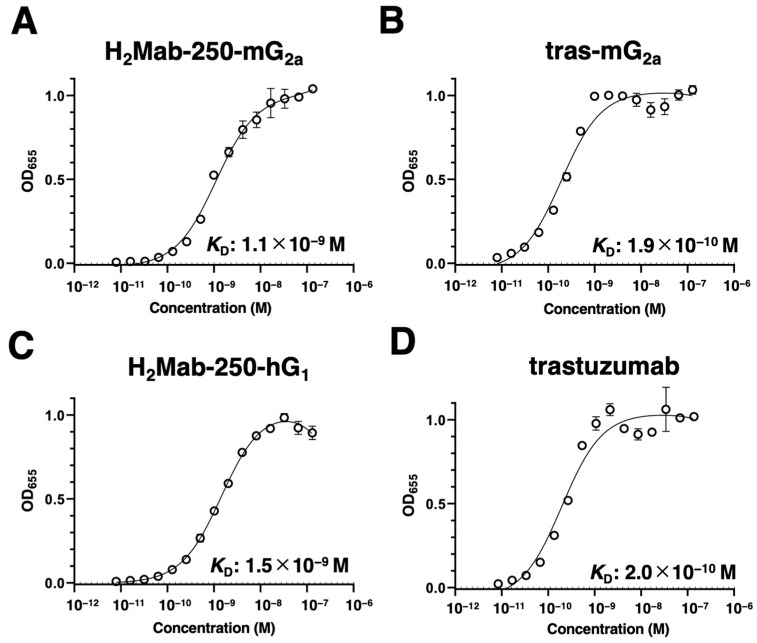
Binding affinity of H_2_Mab-250-mG_2a_, H_2_Mab-250-hG_1_, tras-mG_2a_, and trastuzumab to HER2ec. (**A**–**D**) HER2ec was immobilized on immunoplates and then incubated with the serially diluted H_2_Mab-250-mG_2a_ (**A**), tras-mG_2a_ (**B**), H_2_Mab-250-hG_1_ (**C**), and trastuzumab (**D**), followed by peroxidase-conjugated anti-mouse or anti-human immunoglobulins (n = 3). Enzymatic reactions were conducted, and the optical density at 655 nm was measured. Values are presented as the mean ± SD. The binding isotherms were fitted into the built-in, one-site binding model in GraphPad PRISM 6 to calculate the binding affinity.

**Figure 3 ijms-25-01941-f003:**
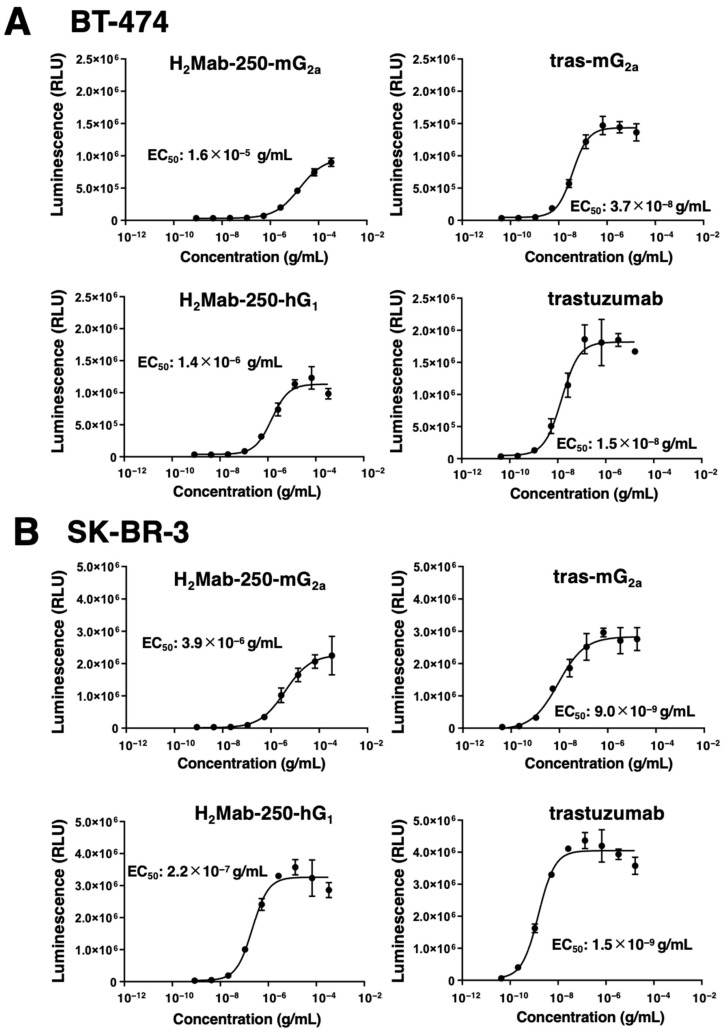
The ADCC reporter assay by H_2_Mab-250-mG_2a_, H_2_Mab-250-hG_1_, tras-mG_2a_, and trastuzumab in the presence of BT-474 and SK-BR-3 cells. Target HER2-positive breast cancer cells such as BT-474 (**A**) or SK-BR-3 (**B**) were cultured in a 96-well white solid plate. H_2_Mab-250-mG_2a_, H_2_Mab-250-hG_1_, tras-mG_2a_, and trastuzumab were serially diluted and added to the target cells (*n* = 3). The engineered Jurkat cells were then added and co-cultured with antibody-treated target cells. Luminescence using the Bio-Glo Luciferase Assay System was measured using a GloMax luminometer (Promega Corporation, Madison, WI, USA). Values are presented as the mean ± SD. The EC_50_ was calculated using GraphPad PRISM 6.

**Figure 4 ijms-25-01941-f004:**
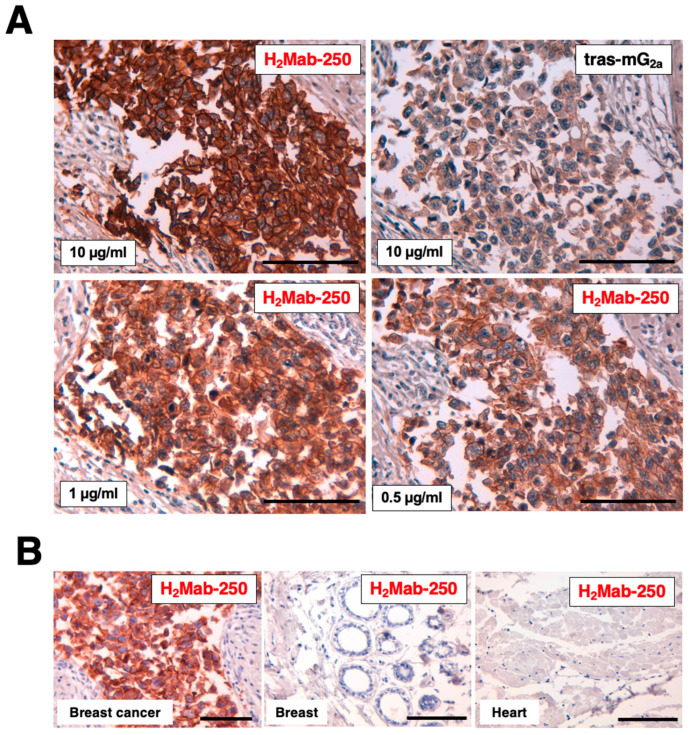
Immunohistochemical analysis of H_2_Mab-250 and tras-mG_2a_ in a breast cancer tissue section and normal tissues. (**A**) HER2-positive breast cancer tissue sections were treated with H_2_Mab-250 (10, 1, and 0.5 µg/mL) or tras-mG_2a_ (10 µg/mL). (**B**) Sections of HER2-positive breast cancer, normal breast, and heart were treated with H_2_Mab-250 (0.1 µg/mL). The sections were then treated with the Envision+ kit (Agilent Technologies, Inc., Santa Clara, CA, USA). The chromogenic reaction was performed using DAB, and the sections were counterstained with hematoxylin. Scale bar = 100 µm.

**Figure 5 ijms-25-01941-f005:**
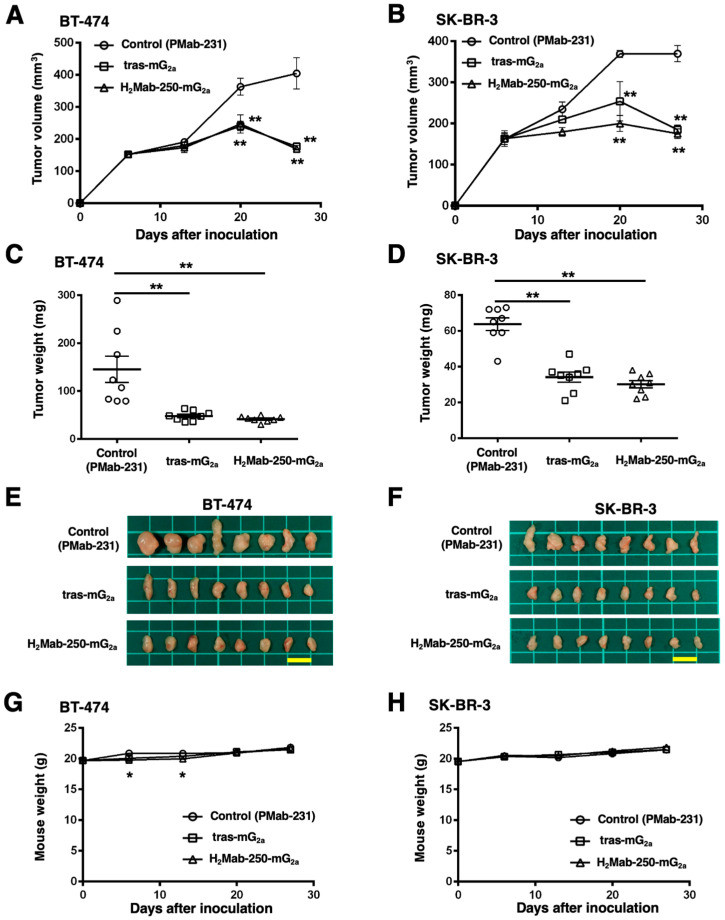
Antitumor activity of H_2_Mab-250-mG_2a_ and tras-mG_2a_ against BT-474 and SK-BR-3 xenografts. (**A**,**B**) BT-474 (**A**) and SK-BR-3 (**B**) cells were injected into BALB/c nude mice (day 0). On day 6, 100 μg of tras-mG_2a_ (*n* = 8), H_2_Mab-250-mG_2a_ (*n* = 8), or a control mAb (PMab-231) (*n* = 8) were injected into mice. On days 13 and 20, additional antibodies were injected. On days 6, 13, 20, and 27, the tumor volume was measured. Values are presented as the mean ± SEM. ** *p* < 0.01 (two-way ANOVA and Tukey’s multiple comparisons test). (**C**,**D**) The tumor weight of BT-474 (**C**) and SK-BR-3 (**D**) xenograft tumors on day 27. Values are presented as the mean ± SEM. ** *p* < 0.01 (ANOVA and Tukey’s multiple comparisons test). (**E**,**F**) The BT-474 (**E**) and SK-BR-3 (**F**) xenograft tumors on day 27 (scale bar, 1 cm). (**G**,**H**) The body weight of BT-474 (**G**) and SK-BR-3 (**H**) xenograft-bearing mice treated with control mAb (PMab-231), tras-mG_2a_, and H_2_Mab-250-mG_2a_. * *p* < 0.05 (two-way ANOVA and Tukey’s multiple comparisons test).

**Figure 6 ijms-25-01941-f006:**
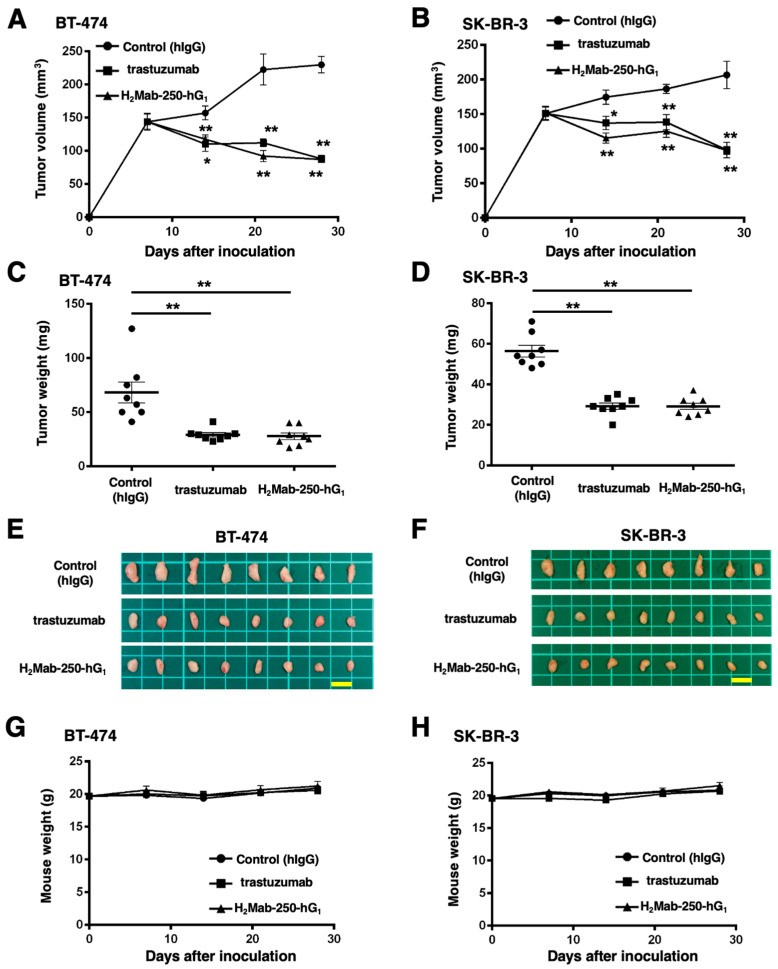
Antitumor activity of H_2_Mab-250-hG_1_ and trastuzumab against BT-474 and SK-BR-3 xenografts. (**A**,**B**) BT-474 (**A**) and SK-BR-3 (**B**) cells were injected into BALB/c nude mice (day 0). On day 7, 100 μg of H_2_Mab-250-hG_1_ (*n* = 8), trastuzumab (*n* = 8), or a control human IgG (hIgG) (*n* = 8) were injected into mice. On days 14 and 21, additional antibodies were injected. Human NK cells were injected around the tumors on the same days of Ab administration. On days 7, 14, 21, and 28, the tumor volume was measured. Values are presented as the mean ± SEM. * *p* < 0.05, ** *p* < 0.01 (two-way ANOVA and Tukey’s multiple comparisons test). (**C**,**D**) The tumor weight of BT-474 (**C**) and SK-BR-3 (**D**) xenograft tumors on day 28. Values are presented as the mean ± SEM. ** *p* < 0.01 (ANOVA and Tukey’s multiple comparisons test). (**E**,**F**) The BT-474 (**E**) and SK-BR-3 (**F**) xenograft tumors on day 28 (scale bar, 1 cm). (**G**,**H**) The body weight of BT-474 (**G**) and SK-BR-3 (**H**) xenograft-bearing mice treated with trastuzumab, H_2_Mab-250-hG_1_, or a control hIgG.

## Data Availability

The data presented in this study are available in the article.

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
