# Peer review of "A Cancer-Specific Monoclonal Antibody against HER2 Exerts Antitumor Activities in Human Breast Cancer Xenograft Models"

_ijms, 2024, doi:10.3390/ijms25031941_

Round 1
Reviewer 1 Report
Comments and Suggestions for Authors
Kaneko and colleagues characterized a defucosylated murine and humanized monoclonal anti-HER2 antibody H2Mab-250/H2CasMab and compared binding and ADCC function in-vitro with an defucosylated trastuzumab antibody. They detected binding with lower affinity and lower effector activity of the H2Mab-250/H2CasMab compared to trastuzumab. In addition, they tested the anti-tumor activity of these antibodies in-vivo using a xenograft model with HER2-pos. BT474 and SK-BR-3 and found comparable efficacy. They postulate that this antibody could be used to generate chimeric antigen receptor-T or antibody-drug conjugates with lower toxicity due to its lower affinity like it has been described for CD19 specific CAR T cells.
However, the group already published the generation of the different antibodies (including the murine H2Mab-250/H2CasMab) and the specific binding to HER2 on cancer but not on normal epithelial cells in a preprint before (Kaneko, M.K.; Suzuki, H.; Kato, Y. Establishment of a Novel Cancer-specific Anti-HER2 Monoclonal Antibody H2Mab-250/H2CasMab-2 for breast cancers. Preprint 2024, doi:10.20944/preprints202309.0906.v5). This preprint also contains information about binding (analyzed by flow cytometry and immunohistochemestry) and function (using the ADCC assay) of 2Mab-250/H2CasMab. The interesting finding like the tumor-specific binding is included in this preprint (but not in the submitted manuscript). The new findings in the submitted manuscript include the analyses of the humanized version of the antibody and in-vivo experiments.
In the last decade, many different anti-HER2 monoclonal antibodies have been generated and characterized before. Moreover, the anti-HER2 antibody trastuzumab is well established in the clinic and different biosimilars have been approved for the clincal application. Trastuzumab has also been used to generate drug conjugates (trastuzumab emtansine (T-DM1) and Trastuzumab-Deruxtecan (T-DXd)), which have been approved for patients and therefore the probability of getting another anti-HER2 +/- conjugate into the clinic are rather low.
Additional remarks:
Unmanipulated trastuzumab antibody should be included in Figure 2: the authors should have tested ADCC activity of the original trastuzumab (not the murine defucolyted version) in comparison with the H2Mab-250/H2CasMabs because this antibody is approved for the clinic and it is difficult to understand why the authors generated a human defucosylated trastuzumab version. The most important scintific question is the advantage of the newly generated antibody in comparison with the approved trastuzumab antibody.
The authors mentioned in the material and method section the information about the human FcγRIIIa ADCC assay, which is adequate for testing antibodies with a human IgG but another assay is available for mouse IgG activity and should be used for the murine antibodies.
The unmanipulated trastuzumab antibody (not the murine defucolyted version) should have been included in the in-vivo studies in Figure 5.
Minor remarks:
1.) the number of samples in each figures should be included in the figure legend
2.) the number of animals in each group needs to be included in the figure legend of Fig. 4 and 5
3.) the "positive" staining of H2Mab-250 in a concentration of 0.1 µg/mL (Figure 3) is not convincing
4.) instead of SKBR-3 and BT474, which are both supposed to be responsive to trastuzumab, the resistant JIMT-1 cell line should be included in the in-vitro experiments
Reviewer 2 Report
Comments and Suggestions for Authors
The manuscript reported a HER-2 positive antibody, H2Mab-250, and investigated its binding affinity, effector activation and in vivo antitumor effect with xenograft models in comparison with trastuzumab. The authors provided a new case of antibody design which trades off target binding affinity for increasing target selectivity. The manuscript overall provided an alternative monoclonal antibody that has comparable antitumor activities than trastuzumab in human breast cancer. I have a few comments below that needs to be addressed before publication.
1. There is no control panel for the comparison of H2Mab-250 and trastuzumab in normal cell line in Figure 1. As the author suggested H2Mab-250 can achieve selectivity by flow cytometry and IHC studies in preprint (https://doi.org/10.20944/preprints202309.0906.v4). H2Mab-250 should show limited binding in normal cells while trastuzumab doesn’t. Because the selectivity data in the preprint manuscript are not peer reviewed, this should be added in current manuscript.
2. A normal cell line staining with H2Mab-250 should be added in Figure 3 due to the same reason as discussed in previous comments.
3. If H2Mab-250 mouse and human derivatives both showing 100 – 1000 times less ability of activating effector cells than trastuzumab-f, why the in vivo antitumor activity is highly similar.
4. The tumor volume data for H2Mab250-mG2a-f in Figure 4A is missing.
5. The format of references such as first letter capitalized is not consistent. Please check the journal publication guideline for formatting.
Round 2
Reviewer 1 Report
Comments and Suggestions for Authors
The authors considerably improved the manuscript by adding the information about tumor specific binding and also redesigned the figures and including addiotional graphs and pictures.
The information about the defucoylation is now eliminated from the whole manuscript – presumably to avoid confusion. This decision is acceptable because details of the generation of the monoclonal antibodies are described in the material and method section.
The information about the clinical trial using this mab as part of a CAR T cell is an important and persuading information and should be included in the manuscript.
Author Response
The information about the clinical trial using this mab as part of a CAR T cell is an important and persuading information and should be included in the manuscript.
>>>
According to the comment, we added the information about the clinical trial using this mAb as part of a CAR T cell.
(Page12, line316)
The property of H2Mab-250 could contribute to the development of HER2-targeting CAR-T cells (now in clinical phase I study in the US) by limiting trogocytosis and maintaining cancer specificity.
Reviewer 2 Report
Comments and Suggestions for Authors
The authors had addressed all my comments and I recommend accepting current manuscript after format adjustment based on journal's guideline.
Author Response
Thank you very much.